# Assessment of Forest Ecosystem Variations in the Lancang–Mekong Region by Remote Sensing from 2010 to 2020

**DOI:** 10.3390/s23229038

**Published:** 2023-11-08

**Authors:** Jing Zhao, Jing Li, Qinhuo Liu, Yadong Dong, Li Li, Hu Zhang

**Affiliations:** 1State Key Laboratory of Remote Sensing Science, Aerospace Information Research Institute, Chinese Academy of Sciences, Beijing 100101, China; zhaojing201483@aircas.ac.cn (J.Z.); liuqh@aircas.ac.cn (Q.L.); dongyd@aircas.ac.cn (Y.D.); lilifs@aircas.ac.cn (L.L.); zhanghu@aircas.ac.cn (H.Z.); 2College of Resources and Environment, University of Chinese Academy of Sciences, Beijing 100049, China

**Keywords:** forest ecosystem, Lancang–Mekong region, FCI, LAI, FVC, GPP, EQI

## Abstract

Five countries in the Lancang–Mekong region, including Myanmar, Laos, Thailand, Cambodia, and Vietnam, are facing the threat of deforestation, despite having a high level of forest coverage. Quantitatively assessing the forest ecosystem status and its variations based on remote sensing products for vegetation parameters is a crucial prerequisite for the ongoing phase of our future project. In this study, we analyzed forest health in the year 2020 using four vegetation indicators: forest coverage index (FCI), leaf area index (LAI), fraction of green vegetation cover (FVC), and gross primary productivity (GPP). Additionally, we introduced an ecosystem quality index (EQI) to assess the quality of forest health. To understand the long-term trends in the vegetation indicators and EQI, we also performed a linear regression analysis from 2010 to 2020. The results revealed that Laos ranked as the top-performing country for forest ecosystem status in the Lancang–Mekong region in 2020. However, the long-term trend analysis results showed that Cambodia experienced the most significant decline across all indicators, while Vietnam and Thailand demonstrated varying degrees of improvement. This study provides a quality assessment of forest health and its variations in the Lancang–Mekong region, which is crucial for implementing effective conservation strategies.

## 1. Introduction

The Lancang–Mekong region in Southeast Asia, which is adjacent to China, comprises five countries: Myanmar, Laos, Thailand, Cambodia and Vietnam. This region is characterized by the flow of the Mekong River, which provides abundant water and forest resources. Existing research in the Lancang–Mekong region primarily focuses on evaluating the impacts of climate and human activities on processes [1,2,3,4,5], as well as assessing the risk of frequent natural disasters [6]. However, forest ecosystems play an irreplaceable role in regulating global carbon balance, mitigating the rise of greenhouse gas concentrations, and maintaining the stability of the global climate [7,8,9,10]. Unfortunately, rapid economic growth in the Lancang–Mekong region has led to significant deforestation due to industrial development, agricultural expansion, and urbanization acceleration [1,2,11]. Especially after a severe drought in Lancang–Mekong region caused by the abnormal climate in 2010 [6], the risks for the forest ecological environment have posed significant challenges to the sustainable development of the Lancang–Mekong region. Therefore, how to quantitatively assess the status of the forest ecosystem and its variations in the Lancang–Mekong region from 2010 to 2020 is an important issue that needs to be addressed.

Remote sensing technology has emerged as the most effective means of obtaining large-scale information and has found widespread application in monitoring and managing forest ecological environments [8,12]. Initially, the normalized difference vegetation index (NDVI) gained popularity as it is sensitive to the greenness of vegetation, allowing for the monitoring of dynamic changes in vegetation on both regional and global scales [13,14]. For forest types, forest coverage is indeed a critical indicator that reflects the abundance of forest resources and the overall state of the ecological balance [15]. As research progressed, additional biophysical variables, such as leaf area index (LAI) [16], fraction of green vegetation cover (FVC) [12,17], and gross primary productivity (GPP) [3], were incorporated to assess the status of the vegetation ecosystem and its long-term temporal variations at a regional and global level. In this research, LAI and FVC consider canopy structures and landscape distributions based on remote sensing observations, while GPP demonstrates carbon desorption capacities. These parameters offer an insight into the vegetation’s health, productivity, and ecological functions compared to solely relying on the greenness of the NDVI parameter.

The analysis of vegetation changes in an ecosystem has evolved from single-parameter approaches to multi-parameter and multi-resolution monitoring. To better understand the complex vegetation ecosystem, a comprehensive ecological quality index based on FVC and NPP parameters for vegetation is used to quantitatively assess the spatiotemporal changes in ecosystem quality over China [18]. Recently, the Ministry of Ecology and the Environment of China issued an ecosystem quality assessment standard, introducing an ecosystem quality index (EQI). This index is calculated based on LAI, FVC, and GPP, and it is designed to offer a comprehensive reflection of the overall condition of vegetation and ecosystems [19,20,21]. It enables a comprehensive assessment of various types of vegetation, including forests, shrubland, grassland, and farmland [20].

The linear regression analysis method is often used to monitor vegetation dynamics; this calculates the slope of a linear relationship between the number of years and a vegetation parameter (NDVI, enhanced vegetation index (EVI), FVC, etc.) for each pixel, assuming that time and vegetation parameter are independent [17,22,23]. Nonlinear regression assumes that the vegetation trends are nonlinear and varied, allowing for it to capture internal abrupt changes and trend shifts [14,24]. Additionally, the Hurst exponent method is often used to predict future vegetation dynamic trends from time series data [25,26]. For the specific purposes of this study, the linear regression analysis method, which provides a straightforward assessment of trends over time, is more suitable for analyzing the vegetation dynamics over a historical time period.

In this study, we conducted an assessment of key vegetation parameters in the Lancang–Mekong region using quantitative remote sensing products, including the forest coverage index (FCI), LAI, FVC, and GPP, from 2010 to 2020. Detailed information about the data products and the specific method employed in this analysis are introduced in Section 2. The results and discussion are presented in Section 3 and Section 4, respectively. The objectives of our research were as follows: (1) to analyze forest health in the year 2020 and understand the trend characteristics of vegetation dynamics from 2010 to 2020 and (2) to identify potential relationships between forest coverage or elevations and key vegetation parameters, aiming to gain a deeper understanding of the factors influencing the forest health trends in the region.

## 2. Materials and Methods

### 2.1. Study Area and Data Resources

#### 2.1.1. Study Area

The study area was located in the Indo-China Peninsula, primarily comprising five countries: Cambodia, Laos, Myanmar, Thailand, and Vietnam, which range from 5°38′ N to 28°33′ N, 81°23′ E to 124°1′ E (Figure 1). The region is characterized by a diverse and complex topography, consisting of mountains and plateaus with altitudes ranging from 1000 m to 3000 m. The dominant vegetation type within the study area is forest, accounting for approximately 45.21%, while cropland and shrubland cover approximately 34.25% and 16.17%, respectively. The climate type of this region is a typical tropical monsoon with high temperatures throughout the year (average temperature above 18 °C) and concentrated precipitation in summer (annual precipitation more than 2000 mm).

#### 2.1.2. Remote Sensing Products

GLC_FC30 product

The latest global 30 m land-cover product, released by the Aerospace Information Research Institute (AIR), Chinese Academy of Sciences (CAS), Beijing, China, covers the period from 2010 to 2020 and has an update frequency of 5 years. This was used to calculate FCI with 500 m spatial resolution in the study area. The GLC_FCS30 product was generated using all Landsat satellite data (Landsat TM, ETM+, and OLI) spanning a period from 1984 to 2020 with a classification system following the benchmark data of 2020, and incorporated a total of 30 surface cover types [27,28,29]. This classification product can be freely downloaded from the data sharing and service portal website (https://data.casearth.cn/en/ (accessed on 2 November 2023)).

2.Global MuSyQ products

The global MuSyQ LAI product, with a spatial resolution of 500 m and a temporal resolution of 4 days from 2010 to 2020, was extracted from Aqua-Terra Moderate Resolution Imaging Spectroradiometer satellite (MODIS) data based on the three-dimensional radiative transfer (3DRT) model for a heterogeneous surface mixed with water and vegetation [30,31]. The global 500 m/4-day MuSyQ FVC product from 2010 to 2020 was estimated using the gap probability theory based on the clumping index and the MuSyQ LAI product [32]. The global 500 m/4-day MuSyQ GPP product from 2010 to 2020 was generated using an improved satellite data-driven light-use efficiency (LUE) model by identifying the potential maximal GPP (GPP_POT_) and the dominant climate control factor for various plant functional types (PFT) [33]. All of these global 500 m/4-day MuSyQ products, including MuSyQ LAI, FVC, and GPP, which were released by AIR, CAS, Beijing, China, can be obtained from the National Earth System Science Data Center at http://www.geodata.cn/ (accessed on 2 November 2023).

3.Climate classification map

The Köppen–Geiger system classifies the climate into five main classes (i.e., tropical (A), arid (B), temperate (C), cold (D), and polar (E) climates) and 30 sub-types on a global scale based on the threshold values and seasonality of monthly air temperature and precipitation [34]. The new global map of Köppen–Geiger climate classification with 1 km spatial resolution from Princeton University, Princeton, United States of America (USA) is freely available for download at www.gloh2o.org/koppen (accessed on 2 November 2023).

### 2.2. Ecosystem Assessment Methods

To quantitatively assess the regional forest health, an ecological assessment was conducted in this study using remote sensing products, as depicted in Figure 2. To ensure compatibility and comparability among various remote sensing products, all data used in this study were resampled to a spatial resolution of 500 m using the nearest-neighbor method and employing the Sinusoidal projection system.

#### 2.2.1. Forest Status Indicators

Based on the global GLC_FCS30 product with a 30 m spatial resolution, the classification of forests, which includes evergreen broadleaved forest (EBF), deciduous broadleaved forest (DBF), evergreen needle-leaved forest (ENF), deciduous needle-leaved forest (DNF), and mixed-leaf forest types (MF), was initially separated from other land cover types. Subsequently, FCI was calculated as the percentage of forests within a spatial resolution of 500 m. Moreover, three indexes (i.e., the annual average LAI, the annual maximum FVC, and the annual cumulative GPP) were used to analyze the forest growth status. The details of these four indicators are listed in Table 1. For areas without any vegetation cover, all four indicators in Table 1 are set to 0. A higher value of the FCI indicates a larger proportion of forest coverage. A higher value of the annual average LAI and the annual maximum FVC indicate better vegetation growth status. A higher value of the annual cumulative GPP indicates a stronger capacity for carbon sequestration by vegetation.

#### 2.2.2. Ecosystem Quality Indicator

In accordance with China’s Ministry of Ecology and Environment [19], the ecosystem quality index (EQI) was constructed using the relative density for three ecological parameters (the annual average LAI, the annual maximum FVC, and the annual cumulative GPP). The relative density is determined by considering the maximum and minimum values of these ecological parameters for different forest types within each Köppen–Geiger climate classification. To achieve this, each of these parameters was normalized to a range of 0–1 using the following equation:(1)$$RVI_{i,j,k}=\frac{F_{i,j,k}-F_{\min i,j,k}}{F_{\max i,j,k}-F_{\min i,j,k}}$$
where $RVI_{i,j,k}$ is the relative density of the ecological parameter *k* in year *i* for climate type *j*; $F_{i,j,k}$ is the value of ecological parameter *k* in year *i* for climate type *j*; and $F_{\max i,j,k}$ and $F_{\min i,j,k}$ are the maximum and minimum values of the ecological parameter *k* in year *i* for climate type *j*, respectively.

We calculated the EQI as follows [19,20]:(2)$$EQI_{i}=\frac{LAI_{i}+FVC_{i}+GPP_{i}}{3}\times 100$$
where $EQI_{i}$ is the ecosystem quality for the *i*th year; $LAI_{i}$ is the relative density of LAI in year *i*; $FVC_{i}$ is the relative density of FVC in year *i*; and $GPP_{i}$ is the relative density of GPP in year *i*.

Based on the ecosystem quality assessment standard of China’s Ministry of Ecology and Environment [19], the value of each EQI classification is shown in Table 2.

#### 2.2.3. Trend Assessment

In this study, the long term trend in vegetation parameters was analyzed using the linear regression analysis method [26]. Therefore, the linear regression relationship between time and the evaluation indicators of vegetation (i.e., the annual average LAI, the annual maximum FVC, and the annual cumulative GPP) was calculated by Equation (3):(3)$$K=\frac{n\times \sum_{i=1}^{n}i\times Temp_{i}-\left(\sum_{i=1}^{n}i\right)\left(\sum_{i=1}^{n}Temp_{i}\right)}{n\times \sum_{i=1}^{n}i^{2}-{\left(\sum_{i=1}^{n}i\right)}^{2}}$$
where *n* represents the number of years, which in this study is 11 (from 2010 to 2020); $Temp_{i}$ refers to the evaluation indicators of vegetation in the *i*th year; and *K* represents the long-term variation trend for each pixel. When *K* > 0, this indicates an increasing long-term variation trend; conversely, when *K* < 0, this suggests a decreasing trend in the evaluation indicator over the same period.

The trend of EQI for the period 2010–2020 (denoted as KEQI) was calculated using Equation (3). To identify different levels of improvement and decreasing trends, classification was derived from the statistical analysis of KEQI values using the significance levels α = 0.05 and 0.1 [24]. Subsequently, KEQI values, separated by ±1.0 and ±0.5, were categorized into five distinct categories, as illustrated in Table 3.

## 3. Results and Analysis

### 3.1. Spatial Distribution of Vegetation Evaluation Indicators in 2020

The spatial distribution of FCI, annual average LAI, annual maximum FVC, and annual cumulative GPP in the Lancang–Mekong region for the year 2020 exhibited similar patterns, with higher values observed in tropical forests and lower values concentrated in nearby urban areas (Figure 3). In the year 2020, in northwestern Myanmar, western Thailand, southeastern Laos, and eastern Cambodia, the FCI exceeded 90%; however, around major cities, the FCI ranged from 10% to 30% (Figure 3a). For areas with an FCI exceeding 90% in Figure 3a, the annual average LAI, annual maximum FVC and annual cumulative GPP were greater than 4, 95%, and 2000 gC/m^2^, respectively (Figure 3b–d). In contrast, for the areas around major cities with FCI less than 30% in Figure 3a, the annual average LAI, annual maximum FVC, and annual cumulative GPP were lower than 1, 80%, and 1500 gC/m^2^, respectively (Figure 3b–d).

In the year 2020, the overall FCI, annual average LAI, annual maximum FVC, and annual cumulative GPP in the Lancang–Mekong region for forest types were 55.73%, 3.41, 93.51%, and 2879.54 gC/m^2^, respectively (Table 4). Myanmar had the highest forest coverage, at 65.93%, followed by Cambodia (59.88%), Laos (54.85%), Thailand (46.43%), and Vietnam (43.81%). However, Laos had the highest annual average LAI (3.99) and annual maximum FVC (95.37%) among the five countries, followed by Myanmar at 3.58 and 94.80%, Vietnam at 3.32 and 93.78%, Thailand at 3.00 and 91.16%, and Cambodia at 2.98 and 90.41%, respectively. Laos also exhibited the highest annual cumulative GPP (3093.06 gC/m^2^), followed by Thailand (2903.91 gC/m^2^), Myanmar (2845.99 gC/m^2^), Vietnam (2824.06 gC/m^2^), and Cambodia (2689.23 gC/m^2^). Combining the information from Figure 3 and Table 4, it can be observed that Laos, ranking third in the FCI parameter, had the highest values for annual average LAI, annual maximum FVC, and annual cumulative GPP. On the other hand, Cambodia, ranking second in the FCI parameter, showed the lowest values for three vegetation evaluation indicators.

Figure 4 presents the spatial distribution of EQI and summary statistics of EQI for each country over the forest ecosystem in the Lancang–Mekong region in 2020. More than 48.47% of the EQI values were higher than 75 (grade I) in 2020; conversely, only 1.03% of the EQI values were lower than 35 (IV and V) around the city with an FCI below 10% (Figure 4a). The overall quality of the forest ecosystem in the Lancang–Mekong region in 2020 was better (EQI = 71.82). Among five countries in the region, Laos had the highest EQI, with a value of 76.88, followed by Myanmar and Vietnam, both with EQI values of 72.45. Thailand had an EQI of 68.79 and Cambodia had an EQI of 67.30 (Figure 4b). In general, the total of grade I (Good) and grade II (Better) combined was greater than 71%. On the other hand, the total of grade IV (Qualified) and grade V (Below standard) combined was lower than 2.5% for each country. In this, Laos stood out with the largest percentage (63.37%) of grade I (Good), showing a high level of annual average LAI, annual maximum FVC, and annual cumulative GPP, consistent with the conclusion in both Figure 3 and Table 4.

### 3.2. Long-Term Trends for Forest Ecosystem Status from 2010 to 2020

#### 3.2.1. Long-Term Trend in Vegetation Evaluation Indicators

Figure 5 presents FCI differences for five countries in the Lancang–Mekong region for the period 2010–2020. Owing to the 5-year update period for fine classification products, the FCIs were extracted in 2010, 2015, and 2020, respectively. The differences in FCI were calculated between 2020 and 2015 (filled left slash in Figure 5) and between 2015 and 2010 (filled cross in Figure 5). The results indicated that the proportion of decreased FCI (Figure 5 azure bars) was higher than that of increased parts for all countries in the Lancang–Mekong region. Moreover, except for Myanmar, the decrease trend in FCI from 2020 to 2015 was stronger than that from 2015 to 2010 for the other countries in the region (Figure 5 red bars).

Figure 6 presents the spatial distribution of the differences in three vegetation evaluation indicators (annual average LAI, annual maximum FVC, and annual cumulative GPP) over the forest ecosystem in the Lancang–Mekong region from 2010 to 2020. The figure indicates that the areas with increasing trends in the three vegetation evaluation indicators were distributed in southeastern Vietnam, northeastern Thailand and western Myanmar. Conversely, the areas with decreasing trends in the three vegetation evaluation indicators were distributed in Laos, Cambodia, and northern Myanmar (Figure 6a–c). Figure 5d provides a summary of three vegetation evaluation indicators from 2010 to 2020 for five countries in the Lancang–Mekong region. The results showed that Cambodia and Laos experienced a higher proportion of decrease trends (75.36%, 66.29%, and 64.08% for Cambodia and 72.39%, 62.33%, and 55.78% for Laos) for the three vegetation evaluation indicators (i.e., annual average LAI, annual maximum FVC, and annual cumulative GPP) compared to increase trends (24.64%, 33.71%, and 35.92% for Cambodia and 27.61%, 37.67%, and 44.22% for Laos). In addition, Myanmar and Thailand had a higher proportion of increase trends for annual maximum FVC (59.12% for Myanmar and 60.57% for Thailand) and annual cumulative GPP indicators (55.98% for Myanmar and 60.54% for Thailand) compared to the annual average LAI indicator (40.70% for Myanmar and 46.20% for Thailand). Among these five countries, Vietnam demonstrated the best recovery trend for the annual average LAI (54.01%) and annual cumulative GPP (66.04%) indicators, and an almost stable trend for the annual maximum FVC indicator (48.57%) (Figure 6d).

#### 3.2.2. Long-Term Variation Trend in EQI (KEQI) from 2010 to 2020

Figure 7 shows the spatial distribution of KEQI and a statistical histogram of KEQI for each country over the forest ecosystem in the Lancang–Mekong region from 2010 to 2020. The overall percentage increase for five countries in the Lancang–Mekong region was 48.95%, the percentage decrease was 45.84% and the percentage unchanged was 5.21%. The areas located in western Myanmar, central Thailand, and northeastern Vietnam showed an improvement trend in KEQI, suggesting positive developments for the health of the forest ecosystem. However, Cambodia demonstrated a significantly deteriorated trend in KEQI, indicating a decline in the overall health and quality of the forest ecosystem (Figure 7a). Among them, the forest health of Cambodia significantly deteriorated from 2010 to 2020, with the highest proportion of decreased areas at 60.34%, followed by Laos at 52.98%, Myanmar at 44.14%, Thailand at 36.24%, and Vietnam at 31.22% (Figure 7b). The proportion of areas in which forest health improved in Vietnam, Thailand, and Myanmar (61.78%, 55.67%, and 47.98%) was greater than that of the areas in which it decreased (33.63%, 39.08%, and 47.02%).

Focusing on the significant deterioration in ecosystem quality in Cambodia from 2010 to 2020 (Figure 7), the regional deforestation in Cambodia was monitored using Chinese Gaofen no. 1 satellite (GF-1) images before 2015 and 2020 (Figure 8). Previous research has suggested that this deterioration is likely due to the deforestation of areas with high forest coverage [35,36,37]. Figure 8a,b illustrates the regional deforestation (central point located in 104.206° E, 14.003° N) caused by urbanization in Khoueng Oddar Meanchey, a province in northwestern Cambodia and Figure 8c,d illustrates the regional deforestation (central point located in 104.672° E, 12.676° N) in Khoueng Kampong Thom, a province in the central part of Cambodia. In Figure 8a,c, the forest is distributed densely in these areas; however, it had been replaced by cities, farmland, and bare land by the year 2020 or 2021 in Figure 8b,d.

### 3.3. Vegetation Evaluation Indicator Variations for Different FCIs

Figure 9a shows the statistics of different forest coverage indices in 2020 for each country in the Lancang–Mekong region. The FCI ranges from 10% to 100% with a 10% interval. The results showed that for each FCI from 20% to 90%, the percentages were relatively low and did not exceed 13% for any of the five countries. Cambodia and Myanmar had higher percentages (37.88% and 35.91%) where the FCI was greater than 90%. On the other hand, Cambodia, Thailand, and Vietnam had notable proportions (17.89%, 27.05% and 20.79%) where the FCI was lower than 10%. Figure 9b shows the difference in FCI between 2020 and 2015, and Figure 9c–e presents statistics on the differences in the annual average LAI, annual maximum FVC, annual cumulative GPP, and KEQI from 2010 to 2020, under different FCIs for each country. The results show that the forest coverage decreased in 2020 compared to 2015 for all intervals, particularly in Cambodia; however, in Thailand, there was a trend toward increasingly dense forest coverage (100%) (Figure 9b). Moreover, the difference in the annual average LAI, annual maximum FVC, annual cumulative GPP, and KEQI for Cambodia and Laos were all below 0 for different FCIs. Cambodia showed more significant decreasing trends compared to Laos in these vegetation evaluation indicators, indicating deterioration in the health of the forest ecosystem (Figure 9c–e). For Myanmar, Thailand, and Vietnam, the annual average LAI, annual maximum FVC, annual cumulative GPP, and KEQI changed from increasing trends to decreasing trends as the FCI increased from 0% to 100%. The amplitude of the variation gradually decreased with the increase in FCI, suggesting that the forest ecosystems were more stable when there was a higher FCI. Although the results for the vegetation evaluation indicators in Thailand and Myanmar showed an increasing trend each year, the amplitude of each vegetation evaluation indicator decreased with the increase in FCI.

### 3.4. The Influence of Different Altitudes on Vegetation Evaluation Indicators

Figure 10 illustrates the statistical data for EQI and KEQI across various elevations for each country within the Lancang–Mekong forest ecosystem region. The elevations range from 250 m to 2000 m with contour intervals of 250 m, and from 2000 m to 3000 m with contour intervals of 500 m. In Figure 10a, the findings indicate a general trend of increasing EQI and FCI as elevation rises, with the exception of Cambodia, where forest coverage is absent above 2000 m. Among the five countries, Cambodia, Laos, and Thailand exhibit forest coverage indices ranging from 50% to 99% at elevations exceeding 500 m. Additionally, their corresponding EQI values surpass 80 at elevations of 500 m, 750 m, and 1250 m, respectively. Conversely, both Myanmar and Vietnam show forest coverage indices below 70% at 1750 m elevation, and the EQI values remain below 81 across all elevations in these two countries. In Figure 10b, the findings indicate negative KEQI values for Cambodia and Laos, implying a decreasing trend as elevation rises. The most significant absolute value, reaching 0.66, was observed at an elevation of 250 m in Cambodia. These results align with those of Figure 9. However, for Myanmar, Thailand, and Vietnam, the KEQI exhibited a decline from 2010 to 2020. Initially displaying positive values, reaching a maximum of 0.46 at an elevation of 250 m, the KEQI gradually transitioned to negative values, reaching a minimum of −0.59 at an elevation of 3000 m. Notably, the KEQI approached zero at elevations of 1500 m or 1750 m.

## 4. Discussion

### 4.1. Uncertainties of This Study

With the increasing accessibility of high-quality quantitative remote sensing products for vegetation parameters, remote sensing has become an indispensable tool for monitoring and assessing forest ecological environments. Utilizing the forest health assessment framework depicted in Figure 2, the indicators, including FCI, annual average LAI, annual maximum FVC, and annual cumulative GPP, at a spatial resolution of 500 m were employed to evaluate the extent of forest coverage, forest quality, spatial distribution, and carbon desorption capacity of the forest ecosystem. Notice that land cover conversions from various forest types (EBF, DBF, ENF, DNF, and MF) to non-forest were taken into account for FCI for the years 2010 and 2015 in this study. Previous study has shown that these conversions represent a non-neglectable effect in the Lancang–Mekong region [3]. But, other indicators and their long-term trends were extracted based on the forest cover in 2020. This study did not take into account the conversion from non-forest areas to forest, because it constituted only a relatively minor aspect.

Furthermore, the EQI calculated from LAI, FVC, and GPP was employed to gauge the overall health of the forest. Considering that the thresholds for LAI, FVC, and GPP were different, all of these three parameters were normalized on a scale ranging from 0 to 1 before calculating the EQI, and then EQI was computed with equal weights [19,20,21]. This standardization was based on the maximum and minimum values observed for different forest types within each climate classification. The only concern is the slight variations in the maximum and minimum values for these three parameters across different years.

### 4.2. Comparative Analysis with Similar Studies

This study primarily focused on examining natural variations in forest health using quantitative remote sensing products, such as LAI, FVC, and CPP. The analysis of these indicators consistently revealed a decline in both the extent of vegetation cover and the overall quality of the forest ecosystem in the Lancang–Mekong region from 2010 to 2020, which is consistent with previous research findings [3,24]. However, it is important to note that deforestation in the Lancang–Mekong region is largely attributed to human activities [35,36,37]. Therefore, future efforts should emphasize the need to incorporate assessments of human activities in order to effectively evaluate forest health.

In this study, EQI, derived from the ecosystem quality assessment standard by the Ministry of Ecology and the Environment of China [19], was calculated using LAI, FVC, and GPP remote sensing products. The indicator EQI provides a comprehensive reflection of the overall state of vegetation and ecosystems, surpassing the limitations of single parameters such as LAI, FVC, or GPP [2,3,16,17]. However, for the complex interactions among the hydrology, geography, and human activities, additional comprehensive evaluation indices, such as the Ecological Environment Pollution Index (EPI), Ecological Environment Management Index (EMI), and Pressure–State–Response (PSR) [38,39], should also be considered for a more holistic assessment of forest health.

Given the importance of forests in maintaining ecological balance, this study focuses on forest types in the Lancang–Mekong region, which comprise over 45% of the total land cover. The results offer valuable insights into forest health in the Lancang–Mekong region over the ten-year period from 2010 to 2020 and explore the relationships between the FCI/elevation and annual average LAI, annual maximum FVC, annual cumulative GPP, EQI, and KEQI. However, croplands and shrublands cover more than 50% of the mainland, while other vegetation types, such as cultivated land and wetlands, also play a significant role in assessing the overall ecological environment [6,40]. Therefore, conservation efforts should extend beyond forests to protect the rich biodiversity of the region, for example, by preserving wetlands, and promoting sustainable cultivated land-use that minimizes harm to the environment.

## 5. Conclusions

This study used key vegetation parameters (including the FCI, annual average LAI, annual maximum FVC, and annual cumulative GPP) based on quantitative remote sensing products to analyze the status of forest growth, and based on the EQI indicator to evaluate forest health in the Lancang–Mekong region from 2010 to 2020. The findings of this study indicate that, in 2020, Laos outperformed Myanmar, Vietnam, Thailand, and Cambodia in a comprehensive evaluation of the forest ecosystem in the Lancang–Mekong region. From the year 2010 to 2020, Cambodia experienced the highest proportion of decline in all indicators related to the forest ecosystem, followed by Laos and Myanmar, while Vietnam and Thailand demonstrated varying degrees of improvement. Furthermore, it was observed that, as the FCI increased, the absolute values of the change rates of the key vegetation evaluation indicators decreased for all five countries, implying that denser forest coverage tended to result in a more stable forest ecosystem. Apart from the impact of natural activities, the deforestation in Cambodia is mostly caused by urbanization and cultivation. Therefore, measures such as limiting human activities and creating forest nature reserves should be implemented to safeguard the forest’s ecological environment in the Lancang–Mekong region.

## Figures and Tables

**Figure 1 sensors-23-09038-f001:**
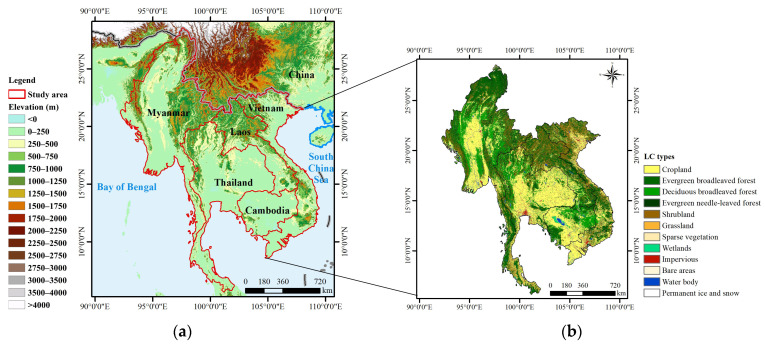
The location and elevation (**a**); and land cover types (**b**) of the study area.

**Figure 2 sensors-23-09038-f002:**
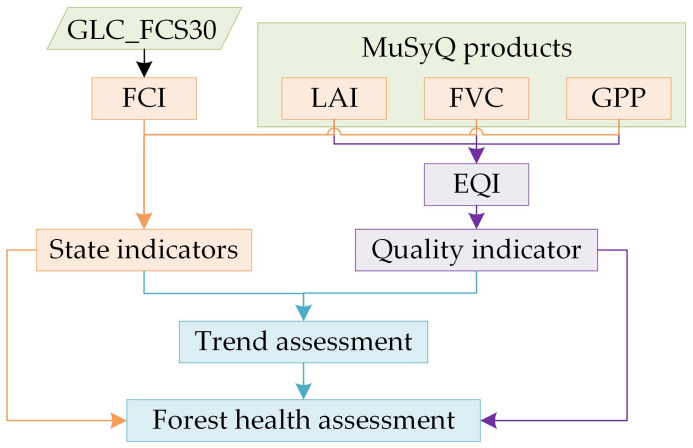
Technical route for forest health assessment based on remote sensing products.

**Figure 3 sensors-23-09038-f003:**
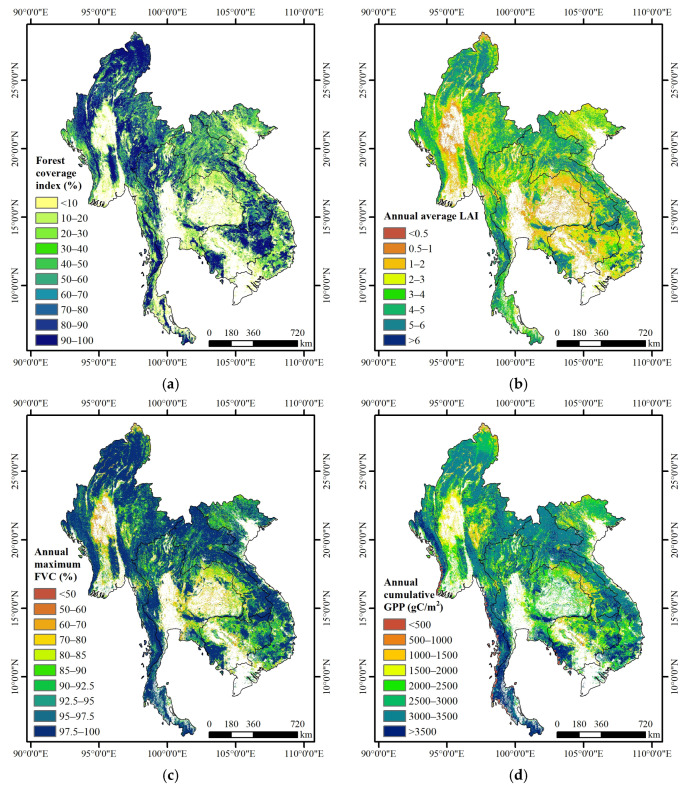
Spatial distribution of vegetation evaluation indicators over the forest ecosystem in the Lancang–Mekong region in 2020. (**a**) FCI; (**b**) annual average LAI; (**c**) annual maximum FVC, and (**d**) annual cumulative GPP (gC/m^2^).

**Figure 4 sensors-23-09038-f004:**
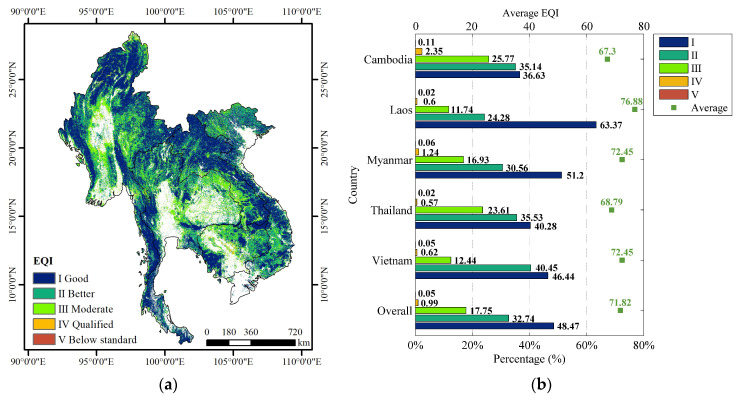
Spatial distribution of EQI for each country in the Lancang-Mekong region in 2020. (**a**) EQI; (**b**) summary of statistics EQI.

**Figure 5 sensors-23-09038-f005:**
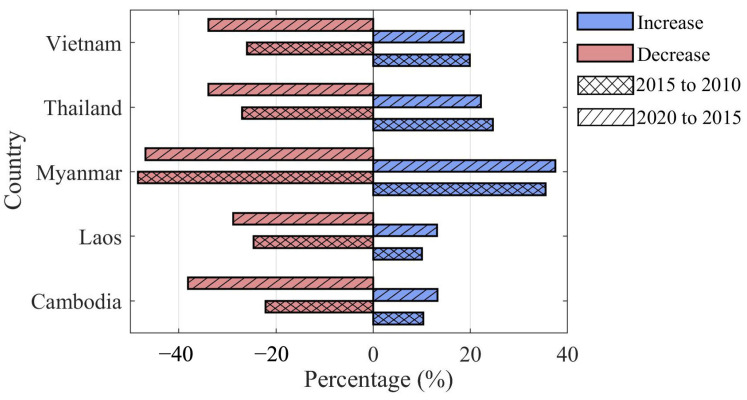
Forest coverage index (FCI) differences from 2020 to 2015, and 2015 to 2010 for five countries in the Lancang–Mekong region. The azure and red colors demonstrate the proportion of increased and decreased areas of FCI, and the filled cross and left slash demonstrate the FCI change from 2015 to 2010 and from 2020 to 2015, respectively.

**Figure 6 sensors-23-09038-f006:**
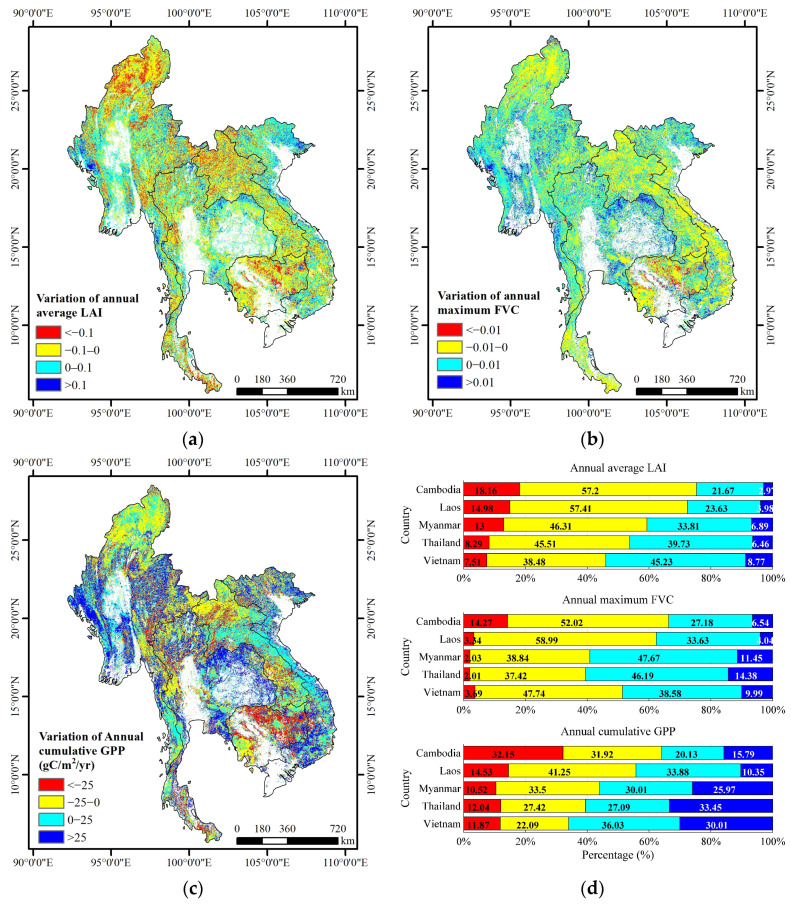
Spatial distribution of long-term variation trends for vegetation evaluation indicators in the Lancang–Mekong region from 2010 to 2020. (**a**) Differences in annual average LAI; (**b**) differences in annual maximum FVC, (**c**) differences in annual cumulative GPP (gC/m^2^/yr), and (**d**) summary of statistics for three vegetation evaluation indicators in five countries. The color classification of red, yellow, cyan, and blue is based on each indicator in (**a**–**c**), respectively.

**Figure 7 sensors-23-09038-f007:**
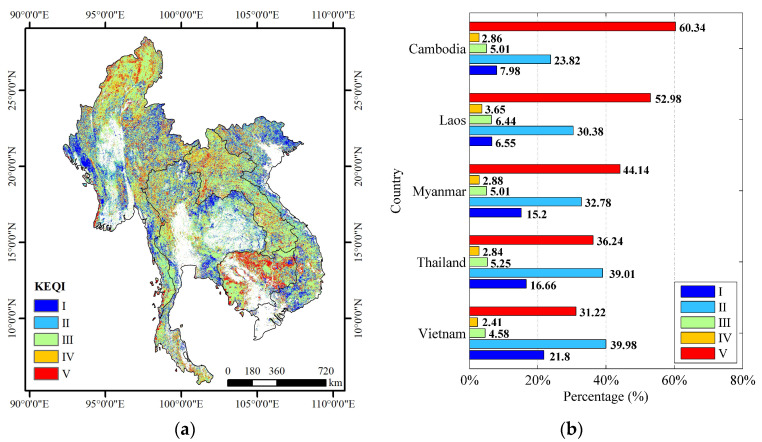
Spatial distribution of KEQI for each country in the Lancang–Mekong region from 2010 to 2020. (**a**) KEQI; (**b**) summary of KEQI statistics.

**Figure 8 sensors-23-09038-f008:**
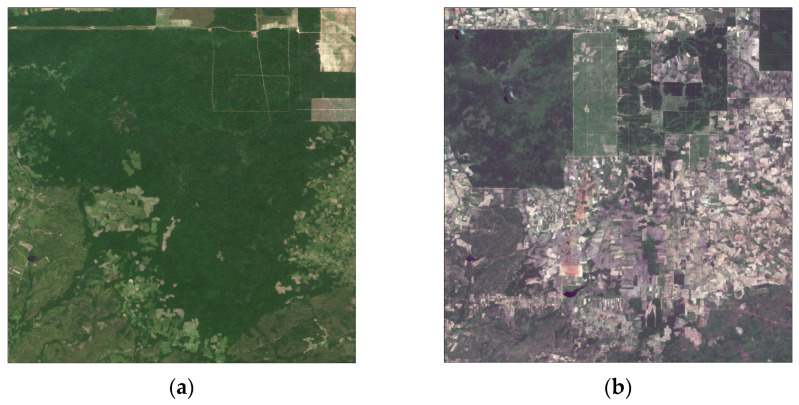
Deforestation of two regions in Cambodia monitored by Chinese GF-1 satellite images with true-color composite (red, green, and blue) before 2015 and 2020, respectively. (**a**) 6 December 2013 in region 1; (**b**) 1 June 2021 in region 1; (**c**) 15 January 2015 in region 2; and (**d**) 24 February 2020 in region 2.

**Figure 9 sensors-23-09038-f009:**
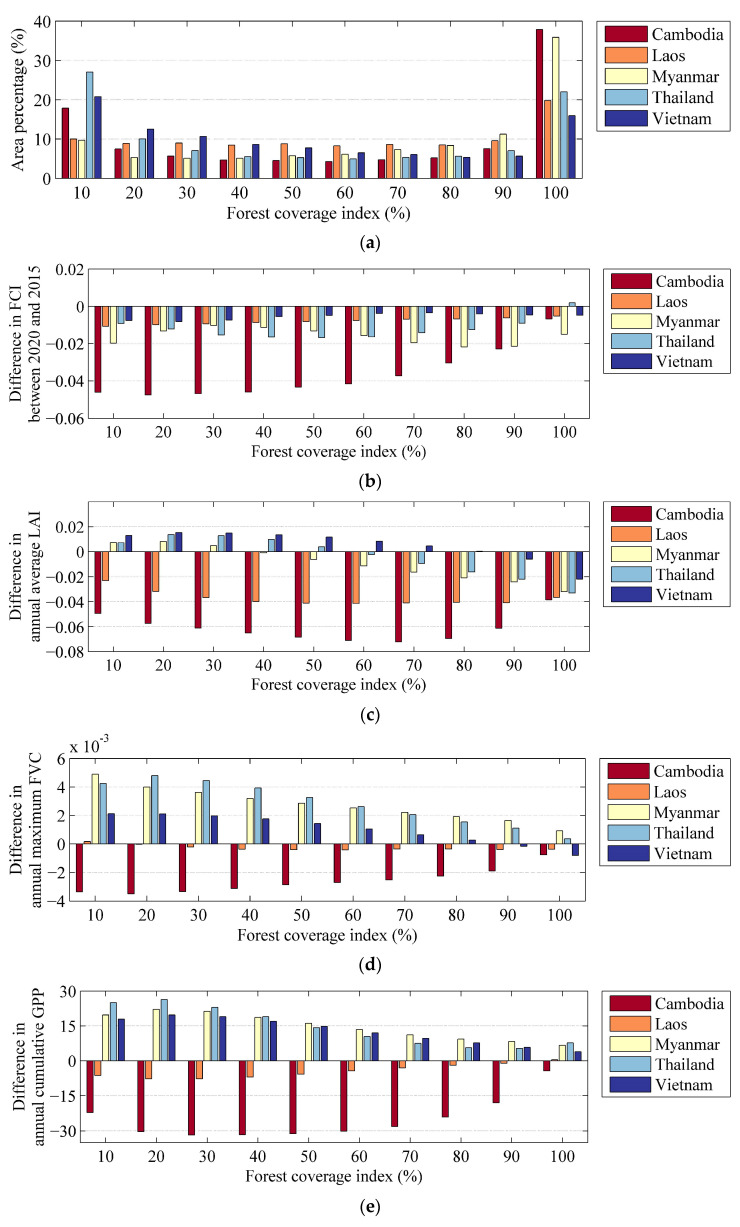
Statistics for the FCI in 2020 and the differences in annual average LAI, annual maximum FVC, annual cumulative GPP, and KEQI from 2010 to 2020 at different FCIs for each country over the forest ecosystem in the Lancang–Mekong region. (**a**) Area percentage of FCI for each country; (**b**) difference in FCI between 2020 and 2015; (**c**) difference in annual average LAI; (**d**) difference in annual maximum FVC, (**e**) difference in annual cumulative GPP (gC/m^2^/yr), and (**f**) KEQI.

**Figure 10 sensors-23-09038-f010:**
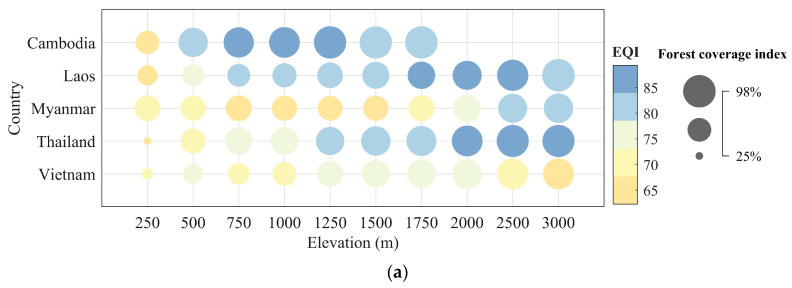
Statistics for EQI in 2020 and statistical histogram of KEQI from 2010 to 2020 at different elevations for each country over the forest ecosystem in the Lancang–Mekong region. (**a**) EQI and FCI for each country; (**b**) variation in KEQI from 2010 to 2020 at different elevations.

**Table 1 sensors-23-09038-t001:** Derails of the forest coverage index (FCI), annual average leaf area index (LAI), annual maximum fraction of green vegetation cover (FVC), and annual cumulative gross primary productivity (GPP) indicating the vegetation growth status for forest ecosystem.

Index	Calculation	Interpretation	Units	Valid Range
FCI	The percentage of forests.	Measure how many forests are distributed in a certain area.	-	[0%, 100%]
Annual average LAI	The average value of LAI for a whole year.	Illustrate the average annual growth of the forest.	m^2^/m^2^	[0, 8]
Annual maximum FVC	The maximum value of FVC during the year.	Display the peak growth over the year.	-	[0%, 100%]
Annual cumulative GPP	The cumulative value of GPP for a whole year.	Demonstrate the carbon sequestration potential of vegetation.	gC/m^2^	[0, 5000]

**Table 2 sensors-23-09038-t002:** Classification Standards for EQI [19].

Grade	I	II	III	IV	V
Description	Good	Better	Moderate	Qualified	Below standard
EQI Range	[75, 100]	[55, 75)	[35, 55)	[20, 35)	[0, 20)

**Table 3 sensors-23-09038-t003:** Classification Standards for KEQI.

Grade	I	II	III	IV	V
Description	Significant improvement	Minor improvement	Unchanged	Slight decrease	Significant decrease
KEQI Range	>1.0/a	(0.5/a, 1.0/a]	[−0.5/a, 0.5/a]	[−1.0/a, −0.5)	<−1.0/a

**Table 4 sensors-23-09038-t004:** Statistics of the forest coverage index (FCI), annual average leaf area index (LAI), annual maximum fraction of green vegetation cover (FVC), and annual cumulative gross primary productivity (GPP) for forest ecosystem in each country in the year 2020.

Country	FCI (%)	Annual Average LAI	Annual Maximum FVC (%)	Annual Cumulative GPP (gC/m^2^)
Overall	55.73	3.41	93.51	2879.54
Cambodia	59.88	2.98	90.41	2689.23
Laos	54.85	3.99	95.37	3093.06
Myanmar	65.93	3.58	94.80	2845.99
Thailand	46.43	3.00	91.16	2903.91
Vietnam	43.81	3.32	93.78	2824.06

## Data Availability

Data used in this study are available on request from the first author.

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
