# Peer review of "Assessment of Forest Ecosystem Variations in the Lancang–Mekong Region by Remote Sensing from 2010 to 2020"

_sensors, 2023, doi:10.3390/s23229038_

Round 1

Reviewer 1 Report

Comments and Suggestions for Authors

The variations of forest ecosystem in the Lancang-Mekong Region were evaluated by using remote sensing index method. The drawings in this manuscript are fairly standard. However, there are still the following major problems to be further clarified by the authors.

1. It is suggested that the author define forest coverage index as the acronym FCI, which is consistent with LAI, FVC and GPP.

2. Why the author chose FCI, LAI, FVC and GPP to evaluate forest ecosystem is still not fully explained in the Introduction. Why not choose NDVI, NPP and other parameters? And do the above four key remote sensing indices have collinearity problems?

3. Authors need to add line numbers to the manuscript to facilitate peer review.

4. The subtitle of Section 2.1.2, "1. Fine classification product" needs to be changed to "Land use product".

5. The spatial resolution and projection coordinates of different remote sensing data sources used by the author are also different. What method does the author adopt to keep the resolution and coordinate system unified?

6. The classification of Table 2 and Table 3 should be supported by references.

7. The discussion part of the paper does not cite any references, which is obviously inappropriate.

8. The analysis results of the paper are not verified. Although the comparison maps of remote sensing images with different phases are shown in Figure 9, such depth of analysis is still insufficient. Please add comparative analysis of relevant studies to the Discussion section to discuss the uncertainty and applicability of this study.

Comments on the Quality of English Language

Some sentences need further polishing.

Author Response

We would like to express our sincere gratitude for your invaluable comments and suggestions, which have undeniably contributed to the enhancement of the quality of our manuscript. The point-by-point response please see the attachment.

Reviewer 2 Report

Comments and Suggestions for Authors

please see comments in the attachment.

Author Response

(The authors gave the same response as above.)

Reviewer 3 Report

Comments and Suggestions for Authors

Please see attached files x2

Comments on the Quality of English Language

Author Response

(The authors gave the same response as above.)

Round 2

Reviewer 1 Report

Comments and Suggestions for Authors

This study used the key vegetation parameters products (including the FCI, annual average LAI, annual maximum FVC and annual cumulative GPP) based on remote sensing data to analyze the forest growth status, and used the EQI indicator to evaluate the forest health in the Lancang–Mekong Basin from 2010 to 2020. However, I do not believe that the current manuscript is ready for publication and further clarification is needed on the following issues:

1. In the calculation equation of EQI (Equation 2), the author regards LAI, FVC and GPP as equal weights, which is obviously inappropriate because it has no clear physical meaning.

2. The classification of KEQI in Table 3 is not supported by literature, which is subjective.

3. Figure 1 requires subgraphs (a) and (b).

4. It is recommended to delete the north finger in the drawing with warp and weft net.

5. In Fig. 4, the mean value of subplot b is instead much larger than the values of the first to fifth categories, which confuses the reader.

6. In the discussion section, the content is too simple, and it is recommended to discuss the following in detail according to the secondary headings: uncertainty of the study; comparative analysis with similar regions, comparative analysis with international peers and cutting-edge research, etc.

7. In the conclusion part, please further delete the content, the text description is a bit lengthy at present.

8. The time span of the author's study is from 2000 to 2020, but the author does not explain the reason for 2010 as the starting year of the study in the introduction. The authors need to explain the background and necessity of 2010 as the starting year of the study.

9. Figure 10 is not appropriate as a discussion section. It is recommended that you place Figure 10 in the results section, or in supplementary materials.

Comments on the Quality of English Language

None.

Author Response

Thank you for the reviewer’s valuable comments. I have attached the point-by-point response here. Please see the attachment.

Reviewer 3 Report

Comments and Suggestions for Authors

Comments on the Quality of English Language

NA

Author Response

(The authors gave the same response as above.)
